# Research on Optimization Design of Cast Process for TiAl Case Casting

**Xiaoping Zhu** [1,2]**, Chunlei Zhu** [2,3]**, Baosen Lin** [2,3] **and Zidong Wang** [1,*]

1   School of Materials Science and Engineering, University of Science and Technology Beijing,
    Beijing 100083, China
2   DEKAI Intelligent Casting Co., Ltd., Zhuozhou 072750, China
3   Beijing Gaona Aero Material Co., Ltd., Beijing 100081, China
*   Correspondence: wangzd@mater.ustb.edu.cn

**Abstract:** The effect of the cast process procedure and pouring systems on several typical metallurgical defects for TiAl case casting was studied by numerical simulation and pouring test. The results indicated that gravity casting had much better melt-filling stability and synchronization, compared to centrifugal casting. Misrun defects in the upper edge of the thin-walled outer ring could be removed by increasing the bottom cross runners, and the negative effect of the overheated zone in the thick-walled flange could be reduced by designing a suitable inner-gate structure. The crack induced by stress concentration in the transition zone brought significant challenges for TiAl case casting and the issue would be effectively resolved through structural design and process optimization.

**Keywords:** TiAl alloys; case casting; gravity casting; centrifugal casting; misrun; crack





## 1. Introduction

Because of their low density, high specific strength, and good creep resistance, TiAl alloys are new lightweight high-temperature structural materials with the potential to replace nickel-based high-temperature alloys in aerospace and marine engine hot-end components [1–4]. TiAl alloys also have higher specific stiffness, higher temperature-specific strength, and lower coefficients of expansion than both nickel-based high-temperature alloys and conventional titanium alloys, making them particularly suitable for making components with high dimensional stability requirements [5,6], such as aircraft engine cases. An aero-engine case usually consists of an outer ring, an inner ring, and support plates connecting the inner and outer rings, with some complex case components having a shunt ring between the inner and outer rings. With an overall diameter that usually exceeds 500 mm and an average wall thickness of 3–5 mm, such components are categorised as medium-to-large thin-walled annular structural parts.

Extensive research on the relationship among the chemical composition—microstructure—mechanical properties and processes of TiAl alloys has been conducted since the 1980s [7,8]. In the 1990s, the American GE company together with Howmet and PCC adopted Ti-48Al-2Cr-2Nb alloy to produce the case components for aircraft engines using a precision casting process, including a radial diffuser ($\varphi$ 610 mm × 62.5 mm, with a minimum wall thickness of 5 mm) and a high-speed civil transport aircraft compressor case. One of these radial diffusers was subjected to engine bench tests [9–11]. It can be inferred that in the US and Europe the development technology TiAl alloy case components has now matured. However, in China, research has focused on small and medium-sized TiAl alloy castings, such as vehicle engine turbocharger turbines [12–14] and aero-engine low-pressure turbine blades [15,16], with no systematic effort to develop medium- and large-sized case castings.

It has been proven that it is very difficult to ensure complete filling and control of the metallurgical quality for case castings made using traditional titanium alloys or nickel-

based high-temperature alloys and even relatively mature casting technologies [17–19]. Compared to titanium and nickel-based high-temperature alloys, TiAl alloys have lower densities, smaller hydrostatic head effects, and melt flows that are lower by an order of magnitude, making filling and metallurgical quality control extremely difficult. Moreover, a welding process to repair TiAl alloy defects has not yet been developed [20], thus, if a casting contains defects over the specification standard, it can only be scrapped. In addition, to date, there have been no studies published on casting methods, casting systems, and process design methods for TiAl case casting. In order to promote TiAl alloy case casting technologies, research on the influence of the casting process and pouring system on the filling behaviour and metallurgical quality of TiAl alloy castings in relation to the structural characteristics of the case and the casting characteristics of the TiAl alloy must be carried out. Taking a case mock-up as the object of study, the research reported in this paper began with a comparative study of the melt-filling behaviour under the centrifugal and gravity pouring processes using ProCAST finite element numerical simulations and pouring test verification. Then, the reasons for the misrun in the upper edge of the case were analyzed, along with the root defect of the thick flange runner and the formation of cracks in the root of the support plates. Finally, solutions were proposed to provide a reference for the casting process and design of the pouring system for TiAl alloy cases.

## 2. Experimental Method

Based on the structural characteristics of a case casting, the mock-up shown in Figure 1 was designed for this study, which consisted of an outer ring, an inner ring, and four support plates connecting the two rings. The outer ring had a maximum diameter of 430 mm and a wall thickness of 8–10 mm. The inner ring had an outer diameter of approximately 200 mm and a wall thickness of approximately 10 mm. The heights of the outer and inner rings were approximately 70 mm. The maximum thickness in the centre of the support plate was approximately 6 mm. The edge of the support plate and root of the support plate were rounded to R2 and R3, respectively. Based on previous experience, the two pouring systems shown in Figure 2 were designed and referred to as 4× and 8× based on the number of cross-pouring runners in the chassis.

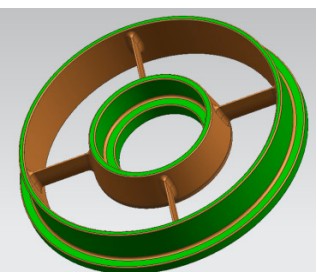

**Figure 1.** Simulation model of case. The out outer diameter of the outer ring is about 430 mm, and the inner diameter of the inner ring is about 200 mm. The average wall thickness is about 8–10 mm. The height of the case is about 70 mm.

ProCAST finite element casting simulation software was used to calculate the melt casting process for the TiAl alloy case castings. A 3D model was created and meshed according to the actual dimensions of the sand mold, runner, and casting, where the side lengths of the mesh cells and runner were 2 mm and 5 mm, respectively. The thermal property parameters used for the main material in the simulations are listed in Table 1. The alloy used for this study was Ti-48Al-2Cr-2Nb (at%) (hereinafter referred to as TiAl). Extensive research has been conducted on Ti-48Al-2Cr-2Nb [21–23]. The mold shell was shaped with magnesia. The pouring temperature in the simulations was 1853 K, and the sandbox pre-heating temperature was 973 K. The casting of the TiAl alloy case was carried out in a vacuum consumable skull furnace with a vacuum level of no more than 1.0 Pa.

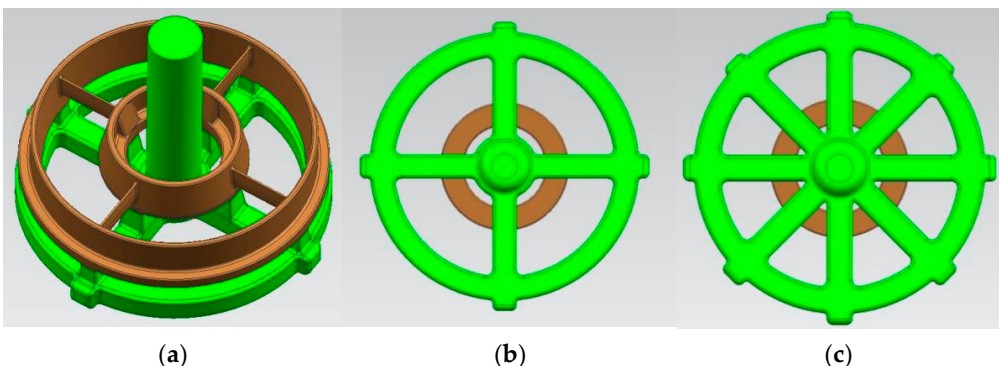

(**a**)                             (**b**)                             (**c**)

**Figure 2.** Gating system (**a**) integrated gating system, (**b**) bottom 4× runner, and (**c**) bottom 8× runner. In (**a**), the plan view, the case is in brown and the gating system is in green. (**b**,**c**) show the bottom view. In (**b**), the metal liquid flows from the 4× cross runner into the case. In (**c**), the metal liquid flows from the 4× cross runner into the case.

**Table 1.** The main thermal and physical properties of TiAl alloy and ceramic shell mold for simulation [24–27].

| Material | Density $(kg \cdot m^{-3})$ | Thermal Conductivity $(W \cdot m^{-1} \cdot K^{-1})$ | Specific Heat $(KJ \cdot Kg^{-1} \cdot K^{-1})$ | Latent Heat $(J \cdot Kg^{-1} \cdot K^{-1})$ | Range of Solidification Temperature (K) | Viscosity Coefficient $(Pa \cdot s \cdot 10^{-3})$ | Interfacial Heat Transfer Coefficient $(W \cdot m^{-2} \cdot K^{-1})$ |
|---|---|---|---|---|---|---|---|
| TiAl | 3900 | 15 [24] | 0.61 [24] | 400 [24] | 1762.5~1781.1 | 4.2~5.4 [25] | —— |
| $Y_2O_3$ shell mold | 4200 | 2.1~2.4 [26] | 0.70~1.00 [26] | —— | —— | —— | 1500 [26] |

## 3. Results and Analysis

### 3.1. Selection of Centrifugal Pouring and Gravity Pouring Process

A TiAl alloy melt has poor fluidity and insufficient filling ability; therefore, it is difficult for TiAl alloys to obtain complete filling, especially for thin-walled parts. To improve the filling ability of the TiAl alloy melt, precision casting and solidification feeding with the help of an external force field is usually carried out in the early stage. For example, Japan Daido Steel Co., Ltd. (Nagoya, Japan) used an anti-gravity field for the development of thin-walled turbocharger wheel castings [28,29], and the China Central Iron and Steel Research Institute adopted a centrifugal casting process for the development of turbocharger wheels [12–14]. The use of an external force field for the development of TiAl alloy castings seems to be necessary. Significantly, gravity casting is also routinely used for conventional titanium case castings. Because TiAl alloy is a special titanium alloy, this study comparatively analyzed the effects of gravity and centrifugal casting on the filling behaviour of TiAl alloy case casting using finite element numerical simulations, in order to provide a reference for the casting processes design of TiAl alloy case.

Figure 3 shows the temperature field evolution of the case castings during the gravity pouring of the 4× gating system. The melt liquid filled horizontally upwards from the bottom runner under the gravitational field, with the outer ring filled simultaneously upwards on all parts of the circumference. In addition, the metal fluid in the support plate area came mainly from the outer ring. The whole melt-filling process was relatively smooth. The melt-filling behavior was essentially the same in all parts of the outer ring. Significantly, the case part studied here was an axisymmetric component, and the melt-filling behavior was the same in the circumferential direction of the outer ring, the four support plates, and the inner ring.

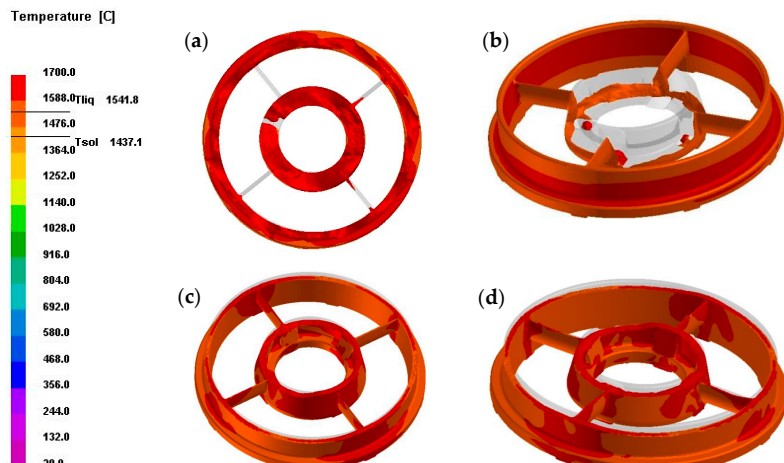

**Figure 3.** Temperature field of melt-filling in the gravity field. The melt liquid was filled horizontally upwards from the bottom runner under the effect of the gravitational field. The whole melt-filling process was relatively smooth. (**a**) the metal fluid in the support plate area came mainly from the outer ring; (**b**)The melt liquid in the four support plates nearly filled from the outer ring into the support plates; (**c**,**d**) the super-heat areas in the outer ring were located near the inner-gate.

Figure 4 shows the temperature field evolution of the case castings during the centrifugal pouring of the 4× gating system at 150 rpm. Under the action of the centrifugal force field, the melt filled the outer ring preferentially. The filling of the parts on the circumference of the outer ring was not synchronised: the outer ring on the preferable side was almost filled while there was still a large area that had not been filled on the other side. As can be seen, the area that had not yet been filled was located between the two water outlets. It is inferred that, if the pouring and pre-heating temperatures of the mold shell were too low, there would be a risk of under-pouring in these areas. For the support plates, the melt entered from the outer ring at 2.64 s. It was found that, unlike gravity pouring, the melt was reversely filled from the outer ring to the support plate under the action of the centrifugal force field. Furthermore, the filling of the four plates was not synchronised, with the melt of the preferential plates reaching the middle of the plate when the melt of the later plates had just entered from the outer ring. The comparison of the gravity and centrifugal pouring processes revealed that, overall, for axisymmetric case castings, the melt-filling behaviour was relatively smooth under gravity pouring conditions, with filling behaviour essentially the same at all equivalent locations. However, under centrifugal pouring conditions, the filling behaviour at the various equivalent positions was not synchronised, and a higher centrifugal speed resulted in a greater tendency for de-synchronisation.

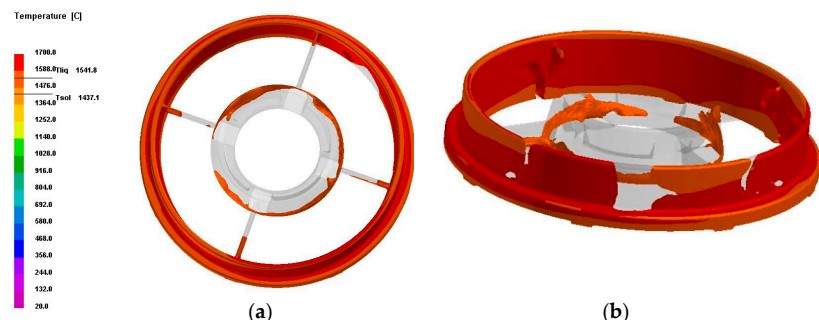

**Figure 4.** Melt mold filling temperature field in the centrifugal force field. The filling behaviour of the melt liquid in four support plates was different. In (**a**), the filling time of the metal liquid in 4 support plates were different. The melt liquid in one support plate has filled into one half while the melt has not filled the other plate. In (**b**), the filling in the outer ring was not synchronised: the outer ring on the preferable side was almost filled while there was still a large area that had not been filled on the other side.

### 3.2. Defect on Upper Edge of Case

Compared with conventional titanium alloys, TiAl has a poorer melt-filling capacity, making the precision casting of thin-walled parts more difficult. As seen in Figure 5, an analysis of the melt flow field showed that, when using gravity pouring for the 4× gating system, the melt filled the outer ring from the bottom flange runner, and the two streams of metal converged in the area corresponding to the middle of the two runners on the upper edge of the outer ring. Under the critical solid phase 5% misrun criterion, the area where the two metal liquids converged had a risk of under-pouring. After an actual casting was poured, under-pouring defects were also found in this area, as seen in Figure 6a,b, showing that the prediction in this study of under-cast defects in the upper edge of the case based on the melt flow field was accurate. Considering that the minimum wall thickness of the outer ring was 8 mm and the height of the outer ring was 70 mm in this study, while the minimum wall thickness is usually less than 5 mm and the height is usually greater than 200 mm for an actual full-size case casting, the precision production of the mock-up case used for testing in this study was less difficult than that of a full-size casting. A significant risk of under-pouring at the upper edge of the full-size casting can also be expected if the gravity pouring process is used.

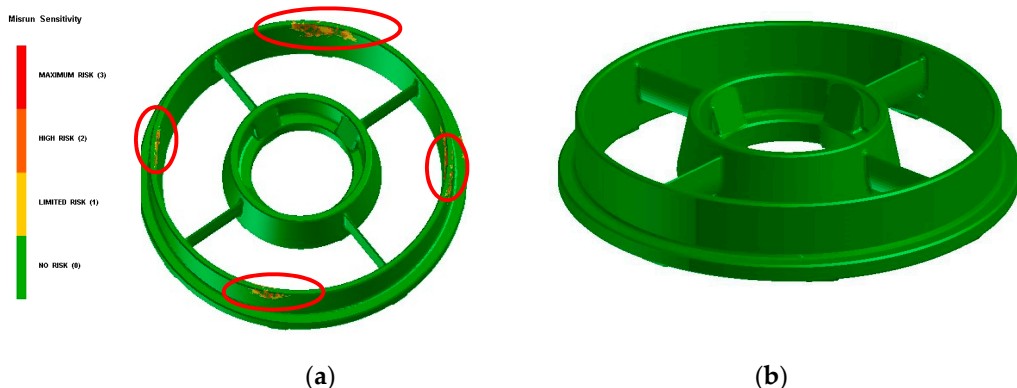

(**a**)            (**b**)

**Figure 5.** Misrun risk prediction results of 4× (**a**) and 8× (**b**) gravity gating systems. In (**a**) misrun risk was seen in the 4× cross runner while misrun risk was not seen in the 8× cross runner.

In order to reduce the risk of under-pouring, this study investigated a proposal to increase the number of bottom runners from four to eight, where the additional cross runners were distributed between two support plates. Finite element calculations of the melt temperature and flow fields showed that there was no risk of under-pouring at the upper edge of the case with the 8× gating (see Figure 5b). After the physical casting of the magazine mock-up, no under-pouring defects were seen on the upper edge (see Figure 6c,d). Significantly, after the number of runners at the bottom was increased, the amount of molten metal in the outer ring of the filling increased, the temperature drops of the molten metal during the first filling decreased, and the melt-filling stroke was shortened by half, all of which helped to improve the melt flow and filling capacity, thereby reducing the risk of under-pouring. Therefore, for full-size TiAl case castings, the number of bottom cross gates needs to be optimized in relation to the actual structural dimensions in order to avoid the problem of under-pouring at the upper edge of the case.

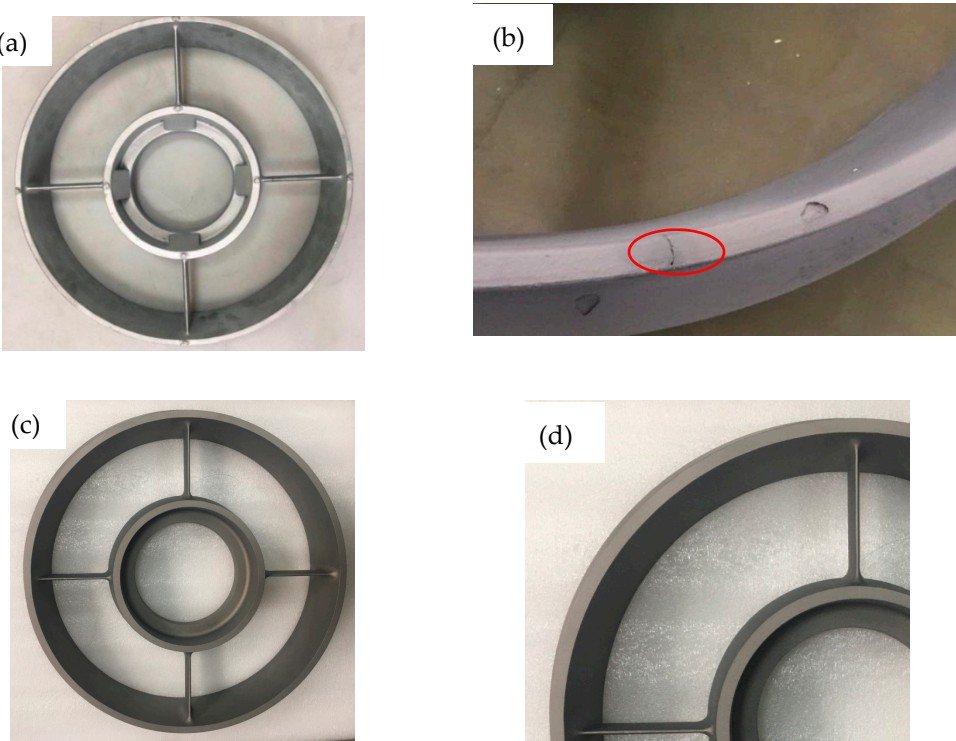

**Figure 6.** The pictures of TiAl cases. Using a 4× cross runner, misrun defects were found in the area corresponding to the middle of the two runners on the upper edge of the outer ring in (**a,b**). Using an 8× cross runner, no misrun defect was found on the upper edge of the outer ring in (**c,d**).

### 3.3. Defects at the Runner of Thick and Large Flange

The cross-sectional flange area of a casting is usually thicker and is therefore the preferred location for internal runners. Conventional titanium cases usually use the flange for the location of internal runners. However, the TiAl alloy melt flow was an order of magnitude lower than that of titanium alloys, and the density was also 10% lower than that of titanium alloys. As a result, the defect-replenishment ability of TiAl castings was significantly reduced, and if the water runner structure is not set up properly, it would be easy for porosity defects to appear at the bottom of the runners. The results of the temperature field calculations showed that for the pouring systems with the 4× and 8× cross runners, there was a degree of overheating at the bottom of the runner (see Figure 7), which indicated a risk of porosity defects there. However, with Niyama's 1% criterion for porosity defects, no porosity defects were seen at the bottom of the runner.

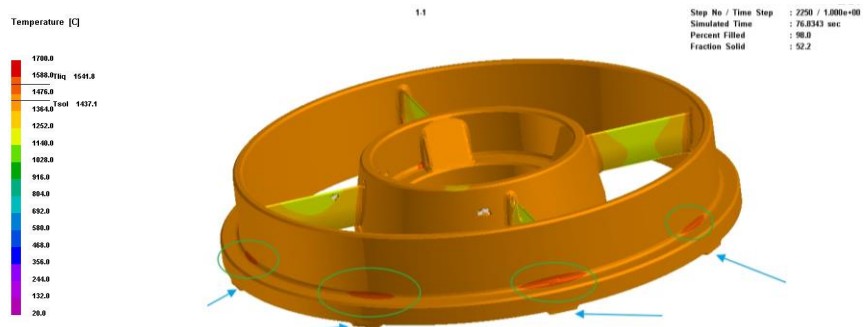

**Figure 7.** Temperature field in the superheated area of the water outlet near the root of the upper water outlet of thick and large flange (arrow points to the water outlet position, and the green circle is the superheated area).

However, after actual pouring, X-ray results showed that there were no porosity defects visible at the bottom of the water runners of the outer and inner rings. However, serious surface defects were found in that area after fluorescence inspection (see Figure 8). and the surface defects were not seen after polishing. Conventional titanium alloy castings need to be pickled to remove the surface $\alpha$-contamination layer [30,31]. The castings investigated in this study were not pickled prior to fluorescence inspection, and, thus, the surface defects may have been related to the surface $\alpha$-contamination layer. Metallographic samples were cut from an area with a serious fluorescent defect at the root of the runner and an area far from the root of the runner where no fluorescent defects were seen for $\alpha$-contamination layer detection. The results showed that the average thickness of the $\alpha$-contamination layer in the area without fluorescent defects was 13 μm, while that in the areas with severe fluorescent defects was approximately 30 μm. It could be inferred that the fluorescence defect at the bottom of the water runner was caused by the surface $\alpha$-contamination layer. Evidently, there was a superheated zone at the bottom of the water runner. Although it did not cause internal porosity defects, the melt reacted violently with the surface of the mold shell during the long solidification time in the superheated zone, thus forming relatively serious surface fluorescence defects. It can be seen that it is necessary to optimise the design of the runner structure at the flange part to reduce the influence of the overheating area at the root of the runner as much as possible. At the same time, it is recommended that the TiAl alloy castings also undergo pickling treatment to remove the surface $\alpha$-contamination layer before fluorescence inspection.

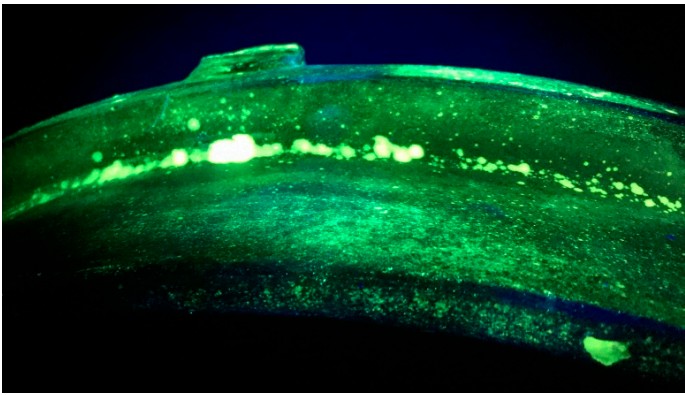

**Figure 8.** Photo of fluorescence detection of an overheated area in casting runner root before polishing and picking. The brighter areas were marked as defects by fluorescence detection.

### 3.4. Stress-Induced Cracks in the Thick-Thin Transition Site

With a room temperature tensile plasticity of less than 2%, which is significantly lower than that of conventional titanium alloys, TiAl is a typical brittle metal material that is highly susceptible to cracking once stress concentrates. Because a technology suitable for repairing defects in TiAl alloy castings has not yet been developed in China or abroad, castings can only be scrapped once cracks appear. It is therefore important to reduce the stress in TiAl alloy castings to avoid crack defects.

The case simulated in this study consisted of thin-walled support plates and relatively thick inner and outer ring flanges. This was a structure that was highly susceptible to stress concentrations at the bottom of the support plates, which could induce cracks in severe instances. Finite element simulation analysis showed that when the temperature in the middle part of the support plates was reduced to 1270 °C, the bottom temperature was 1491 °C, which was still in the liquid-solid phase zone, and the temperature gradient between the middle and bottom was 221 °C/28 mm, or 7.8 °C/mm. After 70 s, the temperatures at the middle and bottom of the support plates were reduced to 1140 °C and 1449 °C, respectively, and the peak temperature gradient between the middle and bottom of the support plates was 10.3 °C/mm. At the same time, considering the rate of reduction of the TiAl alloy linear expansion coefficient ($0.3 \times 10^{-6}$/100 °C), it could be assumed that

the large temperature gradient at the middle of the support plates and root of the support plates would produce a tremendous stress concentration. Because of the insufficient ability of the TiAl alloy to relieve stress concentrations, a very large stress concentration at the root of the support plate would induce root cracking. During the trial production of a TiAl alloy case casting with a diameter of 500 mm, cracks appeared at the root of the support plate after the mold shell had been cleaned (see Figure 9).

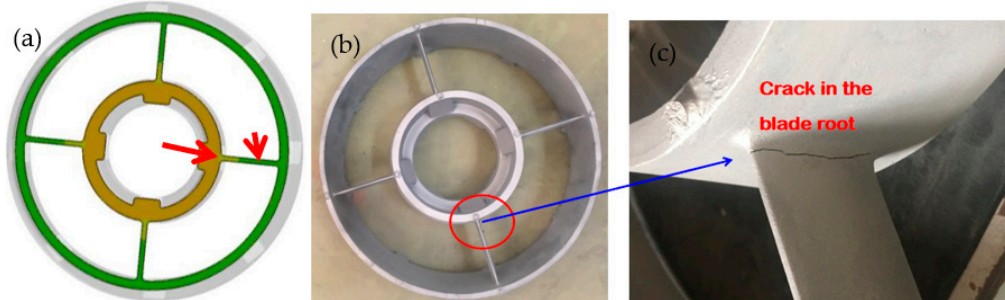

**Figure 9.** Crack at the root of the supporting plate of TiAl alloy case simulation. (**a**) temperature field under 8× cross runner, (**b**) photos of the actual TiAl case, and (**c**) a crack found in the blade root.

## 4. Discussion

Case components are essential parts of an aircraft engine. The application of lightweight TiAl alloys to aero-engine case castings would have a significant structural weight reduction effect. However, the poor fluidity, lack of filling capacity, and high susceptibility to misrun defects of the TiAl alloy melt, along with the lack of applicable defect repair technology, have posed great challenges to the development of TiAl alloy case castings. In this study, a combination of finite element numerical simulations and actual pouring verification was used to investigate the metallurgical quality improvement of TiAl alloy case castings.

This paper first compares the suitability of centrifugal and gravity pouring for TiAl alloy castings. There is no doubt that solidification and filling under the action of external force fields, including centrifugal casting, could significantly increase the filling capacity of TiAl alloy melt, and achieve the precision casting of some thin-walled TiAl alloy cases. For example, the Japan Daido Co. has prepared a civil vehicle engine turbocharger turbine with a minimum wall thickness of only 0.3 mm using anti-gravity casting [25]. The China Central Iron and Steel Research Institute has also prepared a turbocharger turbine with a minimum wall thickness of approximately 1 mm at a diameter level of 100 mm using centrifugal casting [29]. However, the application of centrifugal casting to case castings also brings many disadvantages. For example, (1) the size of the case castings is usually larger, requiring higher dimensional accuracy, and the centrifugal force is greater, which requires the mold shell to have a higher strength in order to avoid problems such as inclusions caused by the mold shell cracking or falling off. Increasing the strength of the mold shell usually reduces the permeability of the mold shell, which in turn increases the risk of porosity-type defects in the castings. (2) The structure of a magazine casting is usually more complex. Under the action of centrifugal force, the melt turbulence increases significantly, which will further increase the risk of melt gas inclusion. (3) Centrifugal casting exacerbates the tendency for the melt to fill asynchronously. This inevitably results in unequal temperature fields for equivalent locations, which in turn causes unequal stress fields and macro- and microstructures. Compared with centrifugal casting, gravity casting requires relatively lower mold shell strength and permeability, and relatively less variation in the melt stability and filling. TiAl alloy case castings can be produced using gravity gating if the pouring system is optimised and a relatively high pouring temperature and mold shell pre-heating temperature are used. The authors have successfully developed a mock-up of a TiAl alloy case ($\varphi$ 420 mm × 70 mm) with qualified metallurgical quality by optimising the pouring system and process.

In the course of this study, it was also found that TiAl alloy cases face the problem of casting stress-induced cracking. This study explored the structural design of castings and the design of the annealing process and achieved certain results. However, actual case castings have thinner walls and more complex structures, with greater stress distributions and influencing factors. Controlling the stress in TiAl alloy case castings is the key to applying TiAl alloy cases in engineering.

## 5. Conclusions

(1) Compared with centrifugal casting, gravity casting has the advantages of melt-filling stability and filling synchronization. With appropriate process parameters, complete filling and metallurgical quality control of TiAl alloy case castings can be realized.

(2) By increasing the number of bottom cross runners to increase the metal liquid filling channel, the melt flow distance is shortened. The mold-filling capacity of TiAl alloy melt to the upper edge of the case is improved, and the misrun risk at the upper edge of the case is reduced.

(3) The thick and large flange part of the TiAl alloy case is prone to porosity defects and surface fluorescence defects due to overheating of the runner. It is necessary to optimize the runner structure to reduce the adverse effects of the overheating zone of the runner.

(4) Stress concentration readily occurs in the thick-thin transition part of TiAl alloy case castings during the casting process. In serious instances, the castings are cracked and discarded. Therefore, various technical measures should be taken to reduce stress concentration and avoid cracks.

**Author Contributions:** Investigation and writing—original draft preparation, X.Z.; simulation analysis, C.Z. and B.L.; Conceptualization, Z.W. All authors have read and agreed to the published version of the manuscript.

**Funding:** This research received no external funding.

**Institutional Review Board Statement:** Not applicable.

**Informed Consent Statement:** Not applicable.

**Data Availability Statement:** Not applicable.

**Conflicts of Interest:** The authors declare no conflict of interest.

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
