# Peer review of "Research on Optimization Design of Cast Process for TiAl Case Casting"

_metals, doi:10.3390/met12111954_

Round 1

Reviewer 1 Report

The manuscript by Zi-dong Wang is a decent paper on casting properties.

In my opinion it can be published after some minor amendments:

1) figures characters are too small impossible to read, Figure 9 has a writing in an unknow language. Chinese ?

2) the English language can be improved.

Reviewer 2 Report

The manuscript needs to be significantly improved / supplemented. 

1. The English needs to be improved. The manuscript should be edited by a native speaker. 

2. The presented results are incomplete, e.g. the simulation results should be related to the microstructure studies.

3. The novelty should be clearly indicated compared to the own published papers and other authors; the research goal should be clearly defined.

4. The captions under most of the figures are incomplete / imprecise and need to be corrected

5. The references need to be updated; more recent works should be cited (only 3 of the 21 items is from 2020/2022);

Reviewer 3 Report

There are multiple recent studies on the TiAl alloys, why don t you refer to them?

Most of your references are older than 5 years!

H.Y. Li, F. Klocke, M. Zeis, T. Herrig, L. Heidemanns,

Experimental Study on the ECM and PECM of Pressed and Casted γ-TiAl Alloys for Aero Engine Applications,

Procedia CIRP,

Volume 68,

2018,

Pages 768-771,

ISSN 2212-8271

Abstract

The Abstract should be written again according to a quality work.

in fact Abstract section should, shortly, should be a summary of the paper, containing the novelty and also some results of the study..

- remove the bad.... from the abstract, you can write the negative effect of...

Line 58-59 you wrote 

 Taking a casing mock-up as the object of study, the research for this thesis

But there is an article, not thesis, isn't  it?

Lines 204-205 you wrote

 The castings investigated in this study were not pickled prior to fluorescence inspection, and the surface defects may have been related to the surface alpha contamination layer..

Therefore, do this procedure before performing the fluorescence analysis!

Where are the X-ray analysis results?

Line 253-257

the following paragraph should be removed, this content is not adequate in a research paper

However, in order to completely solve the problem of cracking at the root of brittle TiAl alloy support plates, the material development and component design departments still need to work closely together to carry out systematic research, including the development of stress control guidelines for thick-to-thin transition areas, root R-angle design guidelines, and stress relief annealing processes. 

In my opinion almost entire manuscript should be reorganized and rewritten according to a quality research paper taking into account all the comments!

Round 2

Reviewer 2 Report

Currently (after correction), the general level of this manuscript and the English language are good and the general reception is positive, however additional proofreading is needed; there are still some structural imperfections to be found and corrected, for example:

1.     Line 22, Keywords: centrifugal casting should be added

2.     Line 218, Fig 6: What is the “…photographes…”;

3.     Line 225,226: The authors written that: "...However, the TiAl alloy melt flow was an order of magnitude lower than that of titanium alloys, and the density was also 10% lower than that of titanium alloys...". Is this statement based on own studies or on the results of other authors? no references;

4.     Line 240,241: The authors written that: “…X-ray results showed that there were no porosity defects visible at the bottom of the water runners of the outer and inner rings…”. Where are the X-ray results?

5.     Fig. 8 and Fig. 9: there are still no marks (a), (b), etc.

6.     …and many other mistakes for corrected;

The simulation results are interesting, but the questions arise, i.e.:

-        Did the authors consider pouring time and solidification time in the numerical simulations?

-       What was the mould filling time in the pouring tests.
